# Non-Traditional Physical Education Classes Improve High School Students’ Movement Competency and Fitness: A Mixed-Methods Program Evaluation Study

**DOI:** 10.3390/ijerph20105914

**Published:** 2023-05-22

**Authors:** Katie M. Heinrich, Cassandra M. Beattie, Derek A. Crawford, Peter Stoepker, Jason George

**Affiliations:** 1Department of Kinesiology, Kansas State University, Manhattan, KS 66506, USA; cbeattie@ksu.edu (C.M.B.); stoepker@ksu.edu (P.S.); 2NDRI-USA, New York, NY 10001, USA; dcrawford@ucmo.edu; 3School of Nutrition, Kinesiology, and Psychological Sciences, University of Central Missouri, Warrensburg, MO 64093, USA; 4Vision Charter School, Caldwell, ID 83605, USA; jasongeorge@visioncsd.org

**Keywords:** CrossFit, weight training, physical education, physical literacy, muscular endurance, strength, aerobic capacity, high-intensity functional training

## Abstract

Non-traditional physical education (PE) programs may facilitate functional movement patterns and develop fitness and work capacity to facilitate long-term physical activity. This program evaluation study compared changes in body composition, movement competency, work capacity, and fitness for high school students in CrossFit or weight training PE; both classes were hypothesized to improve each area, with greater improvements in the CrossFit class. Students participated in 57 min classes 4 days per week for 9 months. Measures including body composition, movement competencies (squat, lunge, push-up, pull-up, hinge, and brace), work capacity (two CrossFit workouts), and fitness (air squats, push-ups, inverted row, plank hold, horizontal and vertical jumps, 5 rep max back squat and press, 500 m bike, and 12 min run) were taken at baseline, midpoint, and post-test. Focus groups to assess students’ experiences and outcomes were conducted at post-test. Students significantly improved in movement competencies (*ps* = 0.034 to <0.001), work capacity (*ps* < 0.001), and all fitness tests (*ps* = 0.036 to <0.001). The CrossFit class was only superior on the 500 m bike. Four themes were identified from the focus groups: (1) increased self-confidence, (2) health improvements, (3) newfound community, and (4) translational sports improvements. Future research should examine changes using an experimental design.

## 1. Introduction

Traditional physical education (PE) programs often emphasize skills and team sports [1]. PE programs focused on sport education at the high school level tend to cater to a particular subset of athletes already involved in those sports [2]. Although sports are important, students must also learn functional movement patterns, address underlying weaknesses and muscle imbalances, and develop the required fitness levels needed to participate in various physical activities [3]. The impacts of a well-designed PE program include improvements in students’ overall fitness markers, psychological outlook, social–emotional health, and cognitive/executive function [2]. Identifying programs that positively impact these variables is critical within the context of the current education system, and these variables are often indicative of a student’s success throughout their education [4]. Physical development during these years is crucial to avoid negative habits that can undermine students’ long-term health and fitness (e.g., injury, metabolic disorders, chronic diseases, etc.) [2].

All students need a basic level of fitness, including both movement competency and work capacity [3]. They can benefit from a fitness protocol that addresses the social camaraderie seen in sports, but also addresses the general physical preparedness (GPP) needed for life [3]. CrossFit has defined its methodology as “constantly varied, high intensity, functional movements” [3]. This type of GPP training offers a comprehensive approach to movement, addresses a variety of energy systems, and leads to well-rounded functional fitness. Previous research indicates the positive impacts of CrossFit across a variety of settings and contexts [5]. Research has also shown CrossFit participation to improve body composition and metabolism while also reducing chronic disease risk factors [6,7]. However, even though CrossFit programs are growing in number within various school districts across the country, only limited research has examined the impacts of CrossFit in a high school setting [8,9,10]. This research has found improvements in aerobic capacity [8], improved self-esteem, perceived body fat and appearance and physical self-concept among those at risk of poor mental health [9], decreased waist circumference and body mass index (BMI) [11], and improved physical fitness [10,11]. However, no research has compared outcomes with another type of PE class.

Muscular power and strength are important components underlying health-related fitness that can be increased through weight training [12]. Weight training utilizes external loads (e.g., barbells, dumbbells, resistance bands, etc.) and even body weight exercises. Adolescents can benefit from weight training programs. For example, nine weight training sessions implemented into PE classes for students in grades 7–9 helped improve the squat and lunge movements for boys and front support brace plus chest touch for girls [13]. However, many PE experiences do not adhere to evidence-based weight training practices and would benefit from delivery by trained and certified professionals [12]. In addition, a lack of suitable equipment for weight training can limit what schools can offer.

The purpose of this study was to examine changes in body composition, movement competency, work capacity, and fitness over an academic school year between high school students participating in CrossFit PE and weight training for athletic performance PE classes. We hypothesized that students in both classes would improve in each measurement with greater improvements for the CrossFit class. The secondary study purpose was to qualitatively examine the students’ experiences in both classes, as they were both different from traditional PE.

## 2. Materials and Methods

### 2.1. Study Setting

Vision CrossFit is a non-profit high school CrossFit affiliate within Vision Charter School, a K–12 Public Charter School in Caldwell, Idaho (i.e., a non-religious tuition-free school that is publicly funded but run independently and bound to a “charter”, including mission, academic goals, and accountability requirements; for more details see https://www.visioncharter.net/about.html; accessed on 15 January 2023). Vision CrossFit was established during the 2016–2017 academic year and this study took place during the 2021–2022 academic year (August–May). Students at Vision CrossFit can enroll in a beginning (Foundations of Fitness and Beginning CrossFit) or advanced (Foundations of Fitness and Advanced CrossFit) course and take this course in place of a traditional physical education course. Students can also take the course for dual credit (i.e., both high school and university) and receive a kinesiology credit in partnership with Northwest Nazarene University, a local University located in Nampa, Idaho.

### 2.2. Study Design and Participants

A mixed-methods program evaluation study was conducted. Invited participants included two classes of 20 students each who had PE class four days per week (i.e., Monday through Thursday) for 57 min per day. Written informed consent was acquired from each student’s parent or guardian and each student provided written informed consent to participate in the study. The study was approved by the Kansas State University Institutional Review Board (#10186; approval date 21 July 2020).

### 2.3. Measures

Program measures were taken on different days over a 3-week time period at baseline (August/September), midpoint (November/December), and post-test (April/May). All in-person assessments were conducted by the class instructor (J.G.) who has degrees in Kinesiology Education (BS) and Science Education (MS), a CrossFit Level 3, Certified CrossFit Trainer certificate, a Precision Nutrition L1, BioForce Certified Conditioning Specialist, and USA Weightlifting Level 2 certification. All assessments were conducted indoors in a temperature-controlled gymnasium. Students were encouraged to perform their best in each assessment and were informed that their performance in each would not impact their grade or class standing in any way.

#### 2.3.1. Body Composition

Directly measured data were used from existing in-class measurements. Height was measured using a Stanley PowerLock tape measure (Towson, MD, USA), and weight was measured using a Weight Watchers digital scale (New York, NY, USA). Those measurements were used to calculate body mass index (BMI) represented as kg/m^2^. Waist circumference was measured with a flexible tape measure.

#### 2.3.2. Movement Competency

Movement competency was measured by assessing students’ ability to complete a body weight squat, forward lunge, modified push-up, modified pull-up, hinge, and brace (modified prone plank) at each timepoint. Standing rotation was assessed only at midpoint and post-test. All assessment protocols and scoring criteria were conducted following the recommendations set forth by Tompsett et al. [14]. The scale for scoring each movement was from 1 (poor) to 5 (desirable). Students were given instruction on the testing procedures prior to each data collection session.

#### 2.3.3. Work Capacity

Two CrossFit benchmark workouts [15] were selected that theoretically represent a range of bioenergetic, motor, and skill demands, representing a broad general work capacity. While all CrossFit workouts contain elements that challenge muscular strength/power and the predominant energy systems, we selected these benchmarks to span the range of aerobic (i.e., Cindy) and anaerobic (i.e., Karen) power capacity. The workout “Cindy” consists of 5 repetitions (reps) of pull-ups, 10 reps of push-ups, and 15 reps of body weight squat exercises. Students had 20 min to complete as many rounds and reps as possible. These rounds and reps were converted to total reps for analysis. The workout “Karen” consists of 150 reps of wallballs (taking a medicine ball through a full squat and throwing it to a target on the wall in front of the individual) performed in as little time as possible. Wallballs were performed with a 6.36 kg medicine ball thrown to a 9-foot target for females and a 9.09 kg medicine ball thrown to a 10-foot target for males. Movements were modified as needed for each student and they were given a 15 min time cap. Then, either the time to complete all reps or the number of reps completed in 15 min was recorded. For analysis, for each rep not completed within the time cap, 3 s was added to the total time per rep.

#### 2.3.4. Fitness Tests

Fitness tests were only conducted at baseline and post-test. To test muscular endurance, students took a single attempt to complete as many repetitions as possible of air squats, push-ups, and inverted row repetitions. They also performed a plank hold for time. To test power, students took three attempts each to achieve their furthest standing horizontal jump and highest vertical jump. For strength, students completed 5 repetitions at the heaviest weight possible for a back squat and 3–5 repetitions for an overhead press. For speed, students cycled 500 m as fast as possible on an air Assaultbike (Carlsbad, CA, USA). To test aerobic endurance, students ran as far as possible in 12 min (i.e., Cooper test). Their total distance was recorded and used to estimate their aerobic capacity with this formula [16]:VO_2_max = (35.97 × miles) − 11.29.

### 2.4. Focus Group

During the last month of school, each class was divided into two groups ranging between 10–12 students, for a total of four focus groups. Efforts were made to include freshmen/sophomores in one focus group and juniors/seniors in the other. Each focus group session was conducted by two researchers via Zoom and took place during their regular class time. Each focus group included the same set of questions that were asked to all participants and additional probing questions were asked for further clarification of responses as needed. A sample question all participants received include, “How has your CrossFit/weight training ability changed this year?” and “Discuss any potential improvements you have experienced over the course of the year?”. Overall, students discussed their CrossFit/weight training experience, CrossFit’s/weight training’s effects on athletic performance, academic performance, executive function, social experiences, and lastly CrossFit/weight training and social norms. The focus groups took 30–45 min and were transcribed verbatim from the Zoom recordings.

### 2.5. Physical Education (PE) Programs

#### 2.5.1. CrossFit PE

The CrossFit class was structured based on the CrossFit Level 2 training guide. After students dressed down for the class, students would meet with the coach at the whiteboard (3–5 min) to discuss the workout of the day, demonstrate new movements, discuss the intended stimulus, and break down an appropriate approach to the workout based on their fitness and/or skill level at the time. In addition, any injuries, potential scaling options, and equipment needed for the workout were discussed. Students were then led through a general warm-up followed by a specific warm-up if needed (8–10 min), depending on the specifics of that workout for the day. Students then performed the workout (5–30 min depending on the stimulus), logged their workouts in the Beyond the Whiteboard app (3 min), and performed a short cooldown and mobility session before dressing down for their next class (3–5 min). Example workouts for 10 days of the class are shown in Appendix A.

#### 2.5.2. Weight Training for Athletic Performance PE

The weight training class was structured like the CrossFit class with some differences primarily pertaining to the workouts chosen (stimulus), which included more traditional strength and conditioning programming that emphasized specific lifts and accessory work over metabolic and energy system training used in the CrossFit class. The weight training class also started each day at the whiteboard to discuss the workout, the specific movement lifts in the workout, any injuries, and intent of the workout. The warm-ups included a general and specific warm-up, but typically the specific warm-up was tailored to the primary lift for that day. For example, if deadlifts were being performed, the specific warm-up would include some high rep deadlifts at lower percentages building to their first working sets. The workouts focused on traditional barbell lifts, such as the squat and its variations, deadlifts, bench and overhead pressing, clean and jerks, and snatches. Accessory work was also programmed with the intent to strengthen weak or lagging muscles and included more bilateral work with dumbbells, kettlebells, and bands. The primary lift typically stayed in the 1–8 rep range, while the accessory lifts stayed in the 8–20 range on a typical day. Students who were interested in competing in Olympic weightlifting were given programming that focused on improving their clean and jerk and their snatch. Students still had time to mobilize via self-myofascial release with the use of rollers, lacrosse balls, and some banded stretching for their cooldown. Example workouts for 10 days of the class are shown in Appendix A.

### 2.6. Statistical Analysis

Data were entered into a spreadsheet by the class instructor (J.G.) and then imported into SPSS version 29 (IBM, Armonk, NY, USA). Means ± SD were computed for all data; chi-square for categorical data and paired-sample t-tests were used to examine baseline differences between classes. Data were screened for normality using the Shapiro–Wilk test [17]. Non-normally distributed data were analyzed with the Wilcoxon Signed Rank test or the Friedman test. Normally distributed data taken at two timepoints were analyzed using 2 (class) × 2 (time) repeated measures ANOVAs and those taken at all three timepoints were analyzed using 2 (class) × 3 (time) repeated measures ANOVAs with post hoc tests using a Bonferroni correction; α was set at 0.05. Partial eta-squared (*η^2^_p_*) was used to estimate effect sizes, interpreted as small (0.01), medium (0.06), and large (0.14).

### 2.7. Focus Group Analysis

Prior to analysis, focus groups were transcribed verbatim by a separate researcher who was not part of the focus group process and transcriptions were imported into Microsoft Word. After focus group transcriptions were completed and verified, two members of the research team independently coded the data into broad themes in a process of open and axial coding [18]. Upon completion of open and axial coding, selecting coding was used and themes were identified.

In order to ensure the reliability of the focus group analysis, two independent coders were utilized to code the database. One coder helped run the interviews and was more familiarized with the data than the second coder, who had to familiarize themselves with the data, allowing for less bias to be present. Themes were also discussed with a member of the research team who was familiar with the data and is an expert in qualitative research through a process of peer-debriefing, and themes were established. Lastly, a negative case analysis was completed to check for any conflicting evidence amongst the identified themes.

## 3. Results

A total of 43 students participated in the study over the 2021–2022 school year at Vision Charter School due to changes in enrollment; 21 were enrolled in CrossFit class and 22 in weight training class. This included 8 freshmen, 11 sophomores, 9 juniors, and 15 seniors. Thirty-five students provided demographic information at baseline, which did not vary significantly between groups. The 19 CrossFit class students were mostly male (68.4%, *n* = 13), mostly white (94.7%), and had an average age of 16 ± 1.0 years. The 16 weight training class students were mostly male (81.3%, *n* = 13), mostly white (67.3%, *n* = 9), and had an average age of 15.6 ± 1.1 years. Overall, one student was black, one Korean, and six students reported being of Hispanic, Latino, or Spanish origin.

### 3.1. Body Composition

Students in the CrossFit class significantly increased height (*Χ^2^* = 6.5, *p* = 0.038) and weight (*Χ^2^* = 6.5, *p* = 0.039) at post-test. Students in the weight training class significantly increased weight (*Χ^2^* = 16.7, *p* < 0.001) and BMI from baseline at midpoint, which was retained at post-test (*Χ^2^* = 16.0, *p* < 0.001). Changes in waist circumference were not significant for either group (see Table 1).

### 3.2. Movement Competency

As shown in Table 2, the CrossFit class significantly improved at each timepoint in movement competencies for squat (*Χ^2^* = 21.0, *p* < 0.001) and modified pull-up (*Χ^2^* = 15.0, *p* < 0.001). They significantly improved from baseline in the lunge (*Χ^2^* = 9.8, *p* = 0.008) and hinge (*Χ^2^* = 13.5, *p* = 0.001). Modified push-up improvements were significantly better at post-test than baseline and midpoint (*Χ^2^* = 8.6, *p* = 0.013). Standing rotation significantly improved from midpoint to post-test (*Ζ* = 3.4, *p* < 0.001).

The weight training class significantly improved at each timepoint in movement competency for the squat (*Χ^2^* = 15.9, *p* < 0.001). They significantly improved at post-test from baseline in the lunge (*Χ^2^* = 11.0, *p* = 0.004) and improved from baseline to midpoint and post-test in the modified push-up (*Χ^2^* = 11.5, *p* = 0.003), modified pull-up (*Χ^2^* = 11.1, *p* = 0.004), and hinge (*Χ^2^* = 8.3, *p* = 0.016). Standing rotation significantly improved from midpoint to post-test (*Ζ* = 2.1, *p* = 0.034).

### 3.3. Work Capacity

There was a significant main effect of time for performance on the 20 min workout Cindy (*ƒ* (2,28) = 18.7, *p* < 0.001), with no significant group X time interaction (see Table 3). The post hoc analysis with a Bonferroni correction indicated that both classes improved their performance at each timepoint. Note that seven participants in the CrossFit class and nine participants in the weight training class completed the workout at all three timepoints.

There was also a significant main effect of time for performance on the workout Karen (*ƒ* (2,16) = 18.4, *p* < 0.001), with no significant group X time interaction. The post hoc analysis with a Bonferroni correction indicated that both classes improved their performance from baseline at midpoint and post-test. Note that five participants per class completed the workout at all three timepoints.

### 3.4. Fitness Tests

Changes in fitness tests are shown in Table 4. The CrossFit class significantly increased their number of air squats (*Ζ* = 3.4, *p* < 0.001) and inverted row reps (*Ζ* = 3.2, *p* = 0.001). The weight training class also significantly increased their number of air squats (*Ζ* = 2.9, *p* = 0.003) and inverted row reps (*Ζ* = 3.1, *p* = 0.002).

The following 2 × 2 RMANOVAs used a Greenhouse–Geisser correction. Both groups significantly improved their push-up reps over time (ƒ (1,29) = 38.2, *p* < 0.001, *η^2^_p_* = 0.57), with no significant differences between groups. Plank hold time also significantly increased within each group (ƒ (1,29) = 7.4, *p* = 0.011, *η^2^_p_* = 0.20), with no significant between-group differences. There was a significant main effect of time for improvements in horizontal jump (ƒ (1,28) = 21.0, *p* < 0.001, *η^2^_p_* = 0.43) and vertical jump (ƒ (1,28) = 87.4, *p* < 0.001, *η^2^_p_* = 0.76) from baseline to post-test. Both groups also significantly increased their 5RM back squat (ƒ (1,19) = 73.8, *p* < 0.001, *η^2^_p_* = 0.80) and 5RM press weights (ƒ (1,11) = 117.0, *p* < 0.001, *η^2^_p_* = 0.91), with no significant between-group differences for either lift. There was a significant group x time interaction for improvements in 500 m bike time, with the CrossFit class improving more than the weight training class (ƒ (1,29) = 4.9, *p* = 0.036, *η^2^_p_* = 0.14), as well as a significant main effect of time (ƒ (1,29) = 16.9, *p* < 0.001, *η^2^_p_* = 0.37). There was a significant main effect of time for 12 min run distance and estimated VO_2max_ (both ƒ (1,28) = 14.9, *p* < 0.001, *η^2^_p_* = 0.35), with no significant between-group differences.

### 3.5. Focus Group Findings

The analysis revealed four prominent themes amongst the focus group data consisting of (1) increased self-confidence, (2) health improvements, (3) newfound community, and (4) translational sports improvements.

#### 3.5.1. Increased Self-Confidence

Students specified improvements in their perceptions of confidence ranging from their class performance to their overall general confidence, with one student in the CrossFit classes reporting, “It helps us build confidence in ourselves”. Another student in the weight training class added to this by saying, “I’d say for me it’s just more of a confidence boost. I feel like I can do more with my life now because I don’t feel limited as much anymore. Definitely helped [sic] bring me out of my shell”. Another student added, “… in weightlifting where it’s like me [sic] if I wasn’t able to like squat something but later in the year I can. It can just kind of give you like the confidence for it and like you know in your head that you have”. Students also shared how confidence in their physical performance increased in both the CrossFit and weight training classes, with a student stating that “…overall, I’ve just gotten a lot stronger and just like more confident that I can actually do the workouts,” and with another mentioning they “…learned to be more confident, even going lower in weight and not comparing myself to others”.

#### 3.5.2. Health Improvements

Students reported that both PE classes had positive impacts on their health, primarily their physical and mental health, with one student in the CrossFit class commenting, “I feel like personally I’ve had my mental and my physical health both have improved”. Another student in the CrossFit PE class noted, “…For some of us I feel like it’s a release from the real-world kind of thing”, while another student noted, “It really is like an outlet, you know, just in everyday it’s like something that you look forward to at the end of the day”. The weight training students had similar responses, with one reporting “I like it. I like how it helps release the emotion”. In addition to mental health, both classes had students report physical health improvements. One CrossFit class student was happy to report, “I have lost a lot of weight and I feel stronger, and I feel like I look better than I have in the past”. Strength was a commonly noted health improvement, with another student in the weight training class reporting, “I feel a lot stronger than I did before”.

#### 3.5.3. Newfound Community

This theme focused on the relationships fostered in both the CrossFit and weight training classes. Students in each class reported that they were able to create special bonds due to the experiences they were able to share together as they progressed through each semester’s workouts. For example, the CrossFit class had one student report the following:

“…I think that like there’s a bond that you make with people in CrossFit that only people in CrossFit know. Like, we’re all suffering like together as a collective group, and so it’s like this. You’re like going through almost like trauma together, and it makes you makes you stronger and like in your friendships and stuff like you can’t go up to somebody to be like, Oh yeah, you remember that workout that just sucked and they like if they don’t know they don’t know like it creates a bond..”..

Another student in the CrossFit class had a similar response, noting that, “To gain bonds, most of us play the same sports together, so having this class and doing the same sports together create more chemistry. So, we just have as like [sic] brothers and sisters in this class..”.. Many students reported building friendships that would have never developed if it were not for their CrossFit or weight training class, while others reported that their pre-existing friendships were able to grow deeper. One weight training student emphasized how they would not have developed the friendships they have today by stating, “There’s definitely a social aspect. Like everyone in this class, I probably wouldn’t have really talked to if I wasn’t in this class, but I’m glad I do because they’re good friends”. Another student in the weight training class had a similar response: “It’s been easy to make friends with people in this class too, because we all share this common interest in weightlifting and suffering”. Another CrossFit class student explained how participating in the classes helped develop deeper friendships between workout partners, stating that without this class, “I don’t think that like we would be as close as we are…we have like a nice bond because of CrossFit..”..

#### 3.5.4. Translational Sports Improvements

Students mentioned they were able to improve in certain areas because their class participation carried over into their sports performance and helped them prepare for competing in their sports. Improvements in endurance, strength, speed, stamina, power, general sport performance, flexibility, and motivation were commonly reported; specifically, endurance was the most frequently reported improvement, followed by strength and sports performance. One student in the CrossFit class reported, “mostly my endurance has gotten a lot better just as the years progressed. Especially basketball, because it’s run up and down the court and I just found myself being like, less out of breath and being able to do it longer,” while a weight training class student reported, “ for me that one’s pretty obvious because I do weightlifting throughout the year, and then I also have basketball so it helped my like [sic] endurance a lot with having two sports”. Strength was noted in both classes, with one weight training class student commenting, “I’ve gotten a lot stronger,” while a CrossFit class student similarly commented, “I’ve gotten stronger. Things have gotten easier. I’ve been able to do more weight on certain things so. It’s just all around helpful for almost anything I do on a daily basis”. Students from both classes noted the carryover from their class into any sport they participate in, with one student from the CrossFit class stating, “I do baseball, basketball, and swimming during the school year and I’ve noticed that, especially with CrossFit it’s helped me just, get better at each so specific sports [sic] and it just helps not just one specific part of each sport, like all around helpful”. Weight training students also reported the general sports improvement carryover from their class, with one saying, “So I’ve always kind of done sports my whole life and I’ve played football for a long time, and I was always kind of heavier-set dude. And before this I could hardly run a mile in under like 15 min. And my athletic ability in sports for sure has gone up. I’m a lot faster and a lot more physical than I used to be”.

## 4. Discussion

This study examined changes in body composition, movement competency, work capacity, sitting time, leisure time, physical activity, and fitness over an academic school year between high school students participating in CrossFit PE and weight training for athletic performance PE classes. Students in the CrossFit class maintained their body composition over the school year, despite increasing height and weight, while students in the weight training class significantly increased both weight and BMI, which may reflect increases in muscle mass. The key finding that fully supported the study hypothesis that both classes would improve with greater improvements for the CrossFit class was that the CrossFit class improved more than the weight training class on the 500 m bike. Other improvements in movement competencies and fitness occurred similarly among students in both classes.

While these classes differed from traditional PE that often emphasizes team sports and related skills [1], they clearly were effective in helping students develop fundamental movement patterns as well as improving fitness, work capacity, and movement competency, all of which support long-term health and fitness [3]. As students performed well on the brace competency at baseline there was no room for significant improvement over time. Interestingly, while the classes varied in focus, the use of the CrossFit template underlying both resulted in similar improvements for students in each class. These results support previous research findings for the positive impacts of CrossFit [5], including improvements in aerobic capacity [8] and physical fitness [10,11]. Given the consensus on each of these forms of training to improve athletic ability, future research should focus more on the behavioral factors and outcomes due to the limited existing literature.

Students in both the CrossFit and weight training classes reported increases in their self-confidence to participate in PE, engagement in physical activity outside of the school day, and overall self-esteem. These findings are consistent with recent evidence demonstrating improved self-perceptions for adolescents participating in CrossFit [9]. Additionally, weight training programs have improved adolescents’ self-efficacy (e.g., confidence in their ability to accomplish a task), self-esteem, self-perceptions, and overall confidence [19,20,21], which was reflected in our findings for increased confidence and self-efficacy. Many students in both classes also reported improvements in both physical and mental health, such as weight loss, muscle mass increases, strength increases, and reductions in stress, which were also reflected in our measured data and concurred with previous CrossFit and weight training research [5,8,10,11,13]. However, only one study has previously shown evidence to suggest CrossFit could improve mental health outcomes of adolescents [9]. Further research should examine the relationship between CrossFit and weight training PE participation by adolescents and positive mental health outcomes.

A supportive community, such as a classroom setting, can provide social support through members showing attention, concern, care, and respect [22]. This concept was reflected in the discussions by students from both classes. Students reported developing a special bond they shared in each class, including increases in sense of community, companionship, and friendship. Although social support is a central part to new and developing communities, there is limited evidence that has examined the impact CrossFit/weight training PE courses can have on positive adolescent community development. CrossFit has been known to foster a special sense of community amongst its participants that not only helps improve their quality of life but also helps aid exercise adherence and overall performance [23,24]. Our study findings not only highlighted the relationships developed in each class, but also highlighted how these relationships helped push the students to perform better, like Lautner at al. [23]. Future research should examine the development of community amongst students in different types of PE classes and the social, physical, and performance variables this might impact. Many of the performance improvements students noted within their class were also reported to carry over into their sports performances, providing evidence for the general physical preparedness model of CrossFit training [3].

Several factors should be considered regarding the implementation of CrossFit PE programs. The specific program evaluated within this study demonstrated many of the key components of implementation success [25]. The Vision CrossFit program was built over a 5-year period and started with limited equipment, including tires, sandbags, homemade sleds (to push or pull), pull-up bars welded to shipping containers, some medicine balls, and jump ropes. This minimal equipment was useful for the teacher to focus on teaching and evaluating movement patterns and correcting movement faults without using much loading (i.e., weights), as well as to work on his creativity in programming workouts. He was able to access do-it-yourself websites that explained how to build workout equipment. In addition, the class did not have a dedicated space and had to work out outside or shared the cafeteria with elementary PE classes. When additional equipment was needed for certain workouts or to have class members compete in the CrossFit Open, local CrossFit affiliates graciously opened their doors to the students. Over time, private donations and funding from the CrossFit Foundation were used to purchase additional equipment. In addition, due to the popularity of the classes, they now have dedicated space within the school. Practitioners who wish to implement CrossFit-based PE need to be aware of the logistical constraints and/or requirements of these programs. Key recommendations are to utilize existing equipment and build up support/demand for the class over time while taking advantage of free resources from CrossFit and then applying for school grants through the CrossFit Foundation.

Strengths of this study include a comparison of CrossFit PE to another unique type of PE class, weight training for athletic performance, delivered by a trained and experienced PE teacher. There was high student compliance during the study, with only students moving in or out of the school changing their participation throughout the year. The mixed-methods nature of the study allowed for a deeper qualitative examination of students’ experiences in both classes over the school year. Limitations included the program evaluation nature of the study, which provides high external but potentially low internal validity due to the potential bias of the teacher conducting the directly measured study assessments. Due to the study design, improvements over the school year may be due to maturation; this could be addressed by randomly assigning students to classes and including a control group. The length of the class (57 min) was challenging to fit everything in, since students had to change clothes for the class each time. In addition, the sample size was small, although statistical significance was found for multiple results.

## 5. Conclusions

This program evaluation study adds to the literature regarding outcomes from alternative types of PE classes among high school students, including CrossFit and weight training. Students in both classes significantly improved in foundational (i.e., basic) human movements, work capacity (i.e., workout performance), and fitness, including muscular endurance (i.e., air squats, inverted row, push-ups, plank hold, horizontal jump, vertical jump, back squat, press, and aerobic capacity). The only significant between-group difference was that the CrossFit class improved significantly more than the weight training class on speed (i.e., 500 m bike). Significant increases in height and weight for the CrossFit class and weight and BMI for the weight training classes may reflect maturation due to the participants’ ages; however, maturation cannot be ruled out for the improvements in work capacity and fitness due to the lack of a control group. Future research should examine students randomly assigned to classes and include a control group.

## Figures and Tables

**Table 1 ijerph-20-05914-t001:** Changes in measured body composition by physical education class type (presented as Mean ± SD).

Variable	CrossFit Class			Weight Training Class		
	Baseline	Midpoint	Post-Test	Baseline	Midpoint	Post-Test
Height (cm)	171.9 (8.0)	171.9 (8.0)	172.9 (7.6) ^2^	175.1 (10.0)	175.1 (10.0)	176.2 (9.0)
Weight (kg)	65.7 (16.8)	65.6 (13.0)	67.0 (14.0) ^2^	74.0 (19.1)	77.1 (19.3) ^1^	78.8 (19.8) ^1^
Body mass index (kg/m^2^)	22.4 (4.8)	22.4 (3.7)	22.5 (3.9)	23.8 (4.5)	24.8 (4.5) ^1^	25.1 (4.8) ^1^
Waist circumference (cm)	77.2 (11.7)	75.7 (10.4)	76.2 (10.3)	83.7 (16.1)	82.6 (13.9)	81.8 (14.6)

^1^ Significantly different from baseline; ^2^ significantly different from baseline and midpoint.

**Table 2 ijerph-20-05914-t002:** Changes in movement competency via basic human movement assessment by physical education class type (presented as Mean ± SD).

Variable	CrossFit Class			Weight Training Class		
	Baseline	Midpoint	Post-Test	Baseline	Midpoint	Post-Test
Squat	3.9 (0.5)	4.5 (0.5)	4.9 (0.3) ^3^	3.9 (0.7)	4.6 (0.5)	4.9 (0.3) ^3^
Lunge	4.5 (0.5)	4.9 (0.4)	5.0 (0.0) ^1^	4.1 (0.8)	4.5 (0.7)	4.9 (0.3) ^1^
Modified push-up	4.5 (0.7)	4.6 (0.5)	5.0 (0.0) ^1,2^	4.2 (0.7)	4.8 (0.5) ^1^	4.7 (0.5) ^1^
Modified pull-up	2.2 (1.6)	4.0 (1.0)	5.0 (0.0) ^3^	2.7 (1.9)	4.3 (1.2) ^1^	4.8 (0.4) ^1^
Brace	4.8 (0.4)	4.9 (0.4)	5.0 (0.0)	4.7 (0.5)	4.8 (0.5)	4.9 (0.3)
Hinge	4.1 (0.5)	4.6 (0.5) ^1^	4.8 (0.4) ^1^	4.1 (0.8)	4.7 (0.5) ^1^	4.7 (0.5) ^1^
Standing rotation	-	4.0 (0.9)	5.0 (0.0) ^2^	-	4.3 (0.6)	4.8 (0.4) ^2^

^1^ Significantly different from baseline; ^2^ significantly different than midpoint; ^3^ significantly different at each timepoint.

**Table 3 ijerph-20-05914-t003:** Changes in work capacity by physical education class type (presented as Mean ± SD).

Variable	CrossFit Class			Weight Training Class		
	Baseline	Midpoint	Post-Test	Baseline	Midpoint	Post-Test
Cindy (repetitions)	336.0 (48.0)	438.7 (36.8) ^1^	524.3 (41.4) ^1,2^	344.3 (42.4)	406.8 (32.5) ^1^	485.7 (36.5) ^1,2^
Karen (mm:ss)	12:56 (3:32)	10:00 (2:45) ^1^	9:33 (4:18) ^1^	12:52 (2:16)	9:04 (1:44) ^1^	9:22 (1:56) ^1^

^1^ Significantly different from baseline; ^2^ significantly different than midpoint.

**Table 4 ijerph-20-05914-t004:** Changes in fitness tests by physical education class type (presented as Mean ± SD).

Variable	CrossFit Class		Weight Training Class	
	Baseline	Post-Test	Baseline	Post-Test
Air squats (reps)	88.8 (73.1)	260.0 (74.5) ^1^	89.4 (66.8)	180.4 (94.3) ^1^
Push-ups (reps)	16.6 (7.4)	32.2 (9.3) ^1^	19.2 (8.9)	27.3 (13.8) ^1^
Inverted row (reps)	4.1 (3.7)	13.8 (4.7) ^1^	4.4 (6.2)	12.0 (7.6) ^1^
Plank hold (seconds)	100.9 (42.8)	151.2 (75.4) ^1^	107.5 (39.4)	147.9 (134.4)
Horizontal jump (cm)	77.6 (11.6)	82.6 (13.7) ^1^	79.3 (10.3)	86.8 (9.7) ^1^
Vertical jump (cm)	19.8 (3.6)	24.1 (3.8) ^1^	20.0 (3.5)	23.6 (4.5) ^1^
5 RM back squat (lbs)	111.7 (38.5)	168.8 (70.7) ^1^	135.6 (43.9)	192.2 (62.9) ^1^
5 RM press (lbs)	58.4 (15.3)	73.6 (18.9) ^1^	70.0 (30.7)	92.5 (30.5) ^1^
500 m bike (ss.h)	42.6 (4.6)	38.2 (3.4) ^1,2^	39.7 (2.7)	38.4 (4.8) ^1^
12 min run (m)	2128.5 (342.4)	2360.3 (444.9) ^1^	1965.4 (301.0)	2253.9 (626.7) ^1^
VO_2max_ (mL/kg/min)	36.3 (7.7)	41.5 (9.9) ^1^	32.6 (6.7)	39.1 (14.0) ^1^

^1^ Significantly improved from baseline; ^2^ significantly greater improvement than the weight training class.

## Data Availability

Data are available upon request from the corresponding author.

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
