# Peer review of "Non-Traditional Physical Education Classes Improve High School Students’ Movement Competency and Fitness: A Mixed-Methods Program Evaluation Study"

_ijerph, 2023, doi:10.3390/ijerph20105914_

Round 1

Reviewer 1 Report

Overall a well-written manuscript.  I only have a few comments for the authors that should be taken care of rather easily.

In the introduction there is a background of sport-education, and the need to focus more on overall fitness.  It seems that this would also be appropriate because it would help children of all abilities improve no matter where they are.  Versus having sport-education which would cater to the students who are already athletic.

In the introduction there is a section on weight training.  While many PE classes don't cover this, could it be the case they don't have the proper equipment, and therefore could not?  That should be addressed. 

In the discussion, I think there should be some part about the feasibility of implementing CrossFit on a larger scale.  Likely, PE teachers would need special training and more work would likely need to go into curriculum, since the workouts change so frequently.  You would also need a tremendous buy in from PE teachers, which could be difficult.  Just thinking of practicality issues.  

Author Response

Overall a well-written manuscript.  I only have a few comments for the authors that should be taken care of rather easily.

Thank you for your positive feedback and helpful comments. We have responded to each below.

In the introduction there is a background of sport-education, and the need to focus more on overall fitness.  It seems that this would also be appropriate because it would help children of all abilities improve no matter where they are.  Versus having sport-education which would cater to the students who are already athletic.

We agree with the reviewer and this is exactly the point we were trying to make in our introduction in lines 44-48: “All students need a basic level of fitness including both movement competency and work capacity [3]. They can benefit from a fitness protocol that addresses the social camaraderie seen in sports, but also addresses the general physical preparedness (GPP) needed for life [3].”

In the introduction there is a section on weight training.  While many PE classes don't cover this, could it be the case they don't have the proper equipment, and therefore could not?  That should be addressed. 

Good point. Equipment can definitely be a limitation. We have added that in the introduction in Lines 67-68 as follows, “In addition, a lack of suitable equipment for weight training can limit what schools can offer.” We have also added information in the discussion per our response to the comment below.

In the discussion, I think there should be some part about the feasibility of implementing CrossFit on a larger scale.  Likely, PE teachers would need special training and more work would likely need to go into curriculum, since the workouts change so frequently.  You would also need a tremendous buy in from PE teachers, which could be difficult.  Just thinking of practicality issues.

We appreciate this suggestion and have added this information in the discussion, lines 430-449.

Reviewer 2 Report

This is an interesting study with great qualitative comments that support the benefits of the chosen non-traditional PE classes. 

I do, however, have a couple of comments related to the study.  Under the Materials and Methods section, you use some labels that might not be familiar with some of the readers of this international journals (ex dual credit & public charter school). I would consider including a brief statement to clarify the meaning of such terms and phrases for those who may not be familiar with certain practices in the US. 

Also, you mentioned baseline data that you collected, were there any other medical evaluations performed?  The participants volunteered for the study, but were they physically fit prior to participating in the study?  

These are a few questions that crossed my mind while reading this manuscript. Otherwise, this is a good study. 

Author Response

This is an interesting study with great qualitative comments that support the benefits of the chosen non-traditional PE classes. 

Thank you for your positive feedback and helpful comments. We have responded to each of your suggestions below.

I do, however, have a couple of comments related to the study.  Under the Materials and Methods section, you use some labels that might not be familiar with some of the readers of this international journals (ex dual credit & public charter school). I would consider including a brief statement to clarify the meaning of such terms and phrases for those who may not be familiar with certain practices in the US. 

These are excellent points. We have added more information about charter schools in lines 79-82. We added an explanation of dual credit in line 87.

Also, you mentioned baseline data that you collected, were there any other medical evaluations performed?  The participants volunteered for the study, but were they physically fit prior to participating in the study?  

Good question. We did not perform any medical evaluations as part of the study, since students were already enrolled in the physical education classes as part of their educational curriculum. The teacher requires the students to complete a Physical Activity Readiness Questionnaire and a health history form, along with a waiver each year. Students who participate in school sports are required to have a physical examination by a physician in grades 9 and 11.

These are a few questions that crossed my mind while reading this manuscript. Otherwise, this is a good study. 

Thank you!